# HsysGNN: Optimizing Distributed Training of Graph Neural Networks in Heterogeneous Systems

## Abstract

With the rapid evolution of GPUs, heterogeneous GPU environments have become increasingly common. However, most existing distributed graph neural network (GNN) training frameworks are designed for homogeneous settings, where discrepancies in GPU performance often exacerbate load imbalance. In this work, we propose a distributed training method tailored for heterogeneous GPU environments. We model GPUs and their interconnects as a computation–communication topology graph, which guides the partitioning of subgraphs such that each GPU is assigned a workload proportional to its computational power and communication bandwidth, thereby achieving balanced utilization across devices. Furthermore, we design a two-level CPU–GPU caching strategy and a pipeline-parallel execution scheme to further reduce inter-partition communication overhead. Experimental results show that, compared with existing approaches, our method significantly improves training performance, while maintaining model accuracy within acceptable bounds and even achieving slight improvements in some cases.

## 1 Introduction

Graph neural networks (GNNs) have become a fundamental tool for learning on relational data, with applications ranging from social network analysis (Sharma et al., 2024) and recommender systems (Wu et al., 2022) to knowledge graphs (Ye et al., 2022). As real-world graphs continue to grow in scale, distributed training across multiple servers and multiple GPUs has become essential.

Existing distributed GNN frameworks have made significant progress in scaling training to large graphs by improving partitioning strategies, caching, and communication efficiency Ma et al. (2019); Jia et al. (2020); Peng et al. (2022); Wang et al. (2022); Lin et al. (2020); Liu et al. (2023); Sun et al. (2023); Yang et al. (2022). However, nearly all of these systems are designed under the assumption of homogeneous GPU clusters, where every device has comparable computational power and interconnect bandwidth, and they assume that each server should be assigned the same number of partitions to ensure balanced distribution. In practice, this assumption rarely holds. Modern clusters are often heterogeneous, due to incremental hardware upgrades, cost-aware deployment in cloud environments, or mixed-use research labs.

Moreove, heterogeneity poses two fundamental challenges. First, *load imbalance*: assigning equal-sized subgraphs to GPUs of unequal speed leads to stragglers, where fast GPUs remain idle while waiting for slower ones. Second, *communication inefficiency*: when inter-GPU links differ in bandwidth (e.g., NVLink vs. PCIe), naively partitioned subgraphs may incur excessive cross-device communication. Together, these issues significantly degrade system throughput and resource utilization, leaving existing distributed GNN frameworks ill-suited for heterogeneous environments.

In this paper, we present HsysGNN, a distributed GNN training framework for heterogeneous GPU systems. HsysGNN models the hardware as a computation–communication topology graph, which guides a hierarchical partitioning scheme that assigns workloads proportional to device capabilities. To further reduce communication overhead, it combines a two-level CPU–GPU caching strategy with pipeline-parallel execution that overlaps computation and communication. Experiments show that HsysGNN delivers up to 18x throughput improvement over state-of-the-art frameworks and 66x

over baseline, while maintaining comparable or even slightly better accuracy. Our contributions are as follows:

- We design a computation–communication topology aware partitioning scheme that assigns workloads proportional to GPU and server capabilities, mitigating stragglers in heterogeneous environments while supporting uneven partitioning across servers.
- We propose a two-level CPU–GPU caching strategy with asynchronous queues to reduce communication bottlenecks and overlap computation with data transfer.
- We provide extensive experiments showing that HsysGNN outperforms existing distributed GNN frameworks under heterogeneous GPU settings, achieving significant speedups while preserving accuracy.

By addressing the challenges of heterogeneity, HSYSGNN brings distributed GNN training closer to the realities of modern computing clusters, where mixed hardware configurations are the norm rather than the exception.

## 2 MOTIVATION

Distributed GNN training relies on graph partitioning to divide computation across devices. In full-batch settings, neighborhood expansion introduces cross-partition dependencies, amplifying both computation and communication demands. However, conventional approaches face inefficiencies in heterogeneous environments:

*Observation 1*: **Straggler Problem in GNN Training.** In a heterogeneous environment, naive equal partitioning leads to load imbalance. Consider a cluster with one fast GPU and one slow GPU: if both are assigned equal subgraphs, the fast GPU finishes early and waits idle for the slow GPU to catch up. This straggler effect wastes computational resources and dominates training time as the degree of heterogeneity increases.

*Observation 2*: **Communication Bottleneck.** If partitioning ignores bandwidth heterogeneity, subgraphs placed on GPUs with weak links (e.g., PCIe) may generate excessive cross-device traffic. Since most GPU clusters route data through CPU memory when peer-to-peer is unavailable, communication latency quickly becomes the bottleneck.

*Observation 3*: **Halo Vertex Overhead.** The ratio of halo to inner vertices grows rapidly with partition count and neighborhood size, leading to redundant memory usage and inflated communication regardless of whether METIS or random partitioning is used.

These observations motivate our design of HSYSGNN, which aims to: (1) balance workload across heterogeneous GPUs by modeling compute capacity in terms of major operators' performance. (2) reduce communication overhead by aligning graph partitioning with interconnect bandwidth. (3) hide data transfer latency through hierarchical caching and pipeline-parallel execution.

## 3 SYSTEM DESIGN

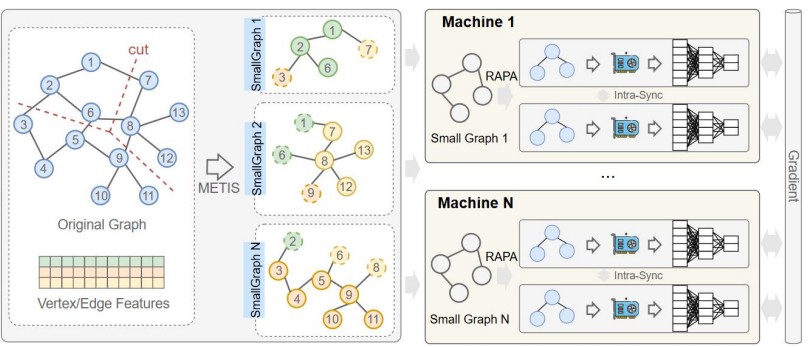

Figure 1: Overall architecture of CaPGNN.

We propose HsysGNN, a distributed GNN training framework explicitly designed for heterogeneous GPU systems. The design is guided by three principles: balancing computation, reducing communication, and hiding latency. Fig. 1 shows the overall architecture.

## 3.1 Computation–Communication Topology Graph

To capture system heterogeneity, HsysGNN models the hardware as a computation–communication topology graph, as shown in Fig. 2, where each GPU is a node weighted by its compute capability and each interconnect is an edge weighted by its effective bandwidth. This abstraction provides a unified view that jointly considers device compute capability and communication heterogeneity.

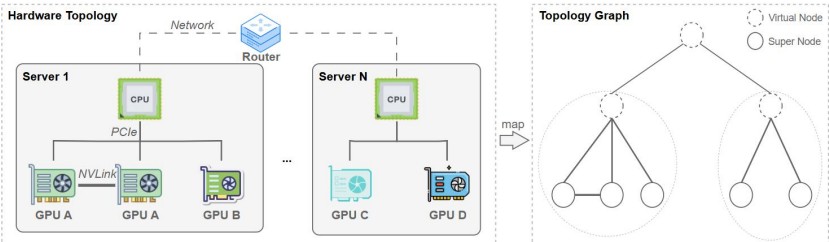

Figure 2: Mapping hardware topology to a computation-communication graph.

The computational workload of GNN training is dominated by two operators: ***message passing***, implemented as sparse–dense matrix multiplication (SpMM) with complexity $O(|E|\cdot d)$, and ***feature transformation***, implemented as dense–dense multiplication (GEMM) with complexity $O(|V|\cdot d^2)$, where $|E|$ and $|V|$ are the number of edges and nodes, $d$ is the feature dimension. To quantify heterogeneous compute capabilities, we measure the operator-level performance of each GPU using microbenchmarks. Specifically, we define $\alpha_i$ as the per-edge cost of SpMM on GPU $i$, and $\beta_i$ as the per-node cost of GEMM. The step cost can then be estimated as,

$$T_i \approx \alpha_i \cdot |E| \cdot d + \beta_i \cdot |V| \cdot d^2 \tag{1}$$

Since different GNNs place different emphasis on SpMM and GEMM, we introduce a fusion metric,

$$C_i = w_\alpha \cdot \alpha_i + w_\beta \cdot \beta_i \tag{2}$$

where $w_\alpha$ and $w_\beta$ reflect model-specific operator weights. For example, $w_\alpha = w_\beta = 0.5$ for GCN, $w_\alpha = 0.7, w_\beta = 0.3$ for GraphSAGE, and attention models such as GAT additionally involve sampled dense–dense multiplication (SDDMM) with cost $O(|E| \cdot d)$, which can be folded into $w_\alpha$. The effective compute capability of GPU $i$ is defined as the reciprocal of this fusion cost, $P_i = 1/C_i$.

In addition to computation, communication heterogeneity plays a central role in distributed training. For each GPU pair $(i, j)$, we measure the effective bandwidth $B_{ij}$ of the interconnect. These measured values are used as edge weights in the topology graph.

By combining fused operator performance with measured interconnect bandwidth, the computation–communication topology graph provides a principled abstraction of heterogeneous systems. This representation allows HsysGNN to assign workloads proportional to GPU capability while minimizing traffic across weak links, laying the groundwork for our partitioning algorithm.

## 3.2 Heterogeneity-Aware Graph Partitioning

Given the computation–communication topology graph, our goal is to assign workloads proportional to GPU capability while avoiding stragglers. Fig. 3 shows HsysGNN achieves this through a hierarchical partitioning strategy: first dividing workloads across servers, and then partitioning within each server.

**Server-Level Partitioning** In multi-server environments, overall performance is bounded by the slowest server due to synchronization. We therefore allocate subgraphs in proportion to each server's aggregate throughput. Specifically, for server $s$ with GPU set $\mathcal{G}_s$, the throughput is estimated as $P_s = \sum_{g \in \mathcal{G}_s} P_g$, where $P_g$ is the compute capability of GPU $g$ derived from the fusion metric in

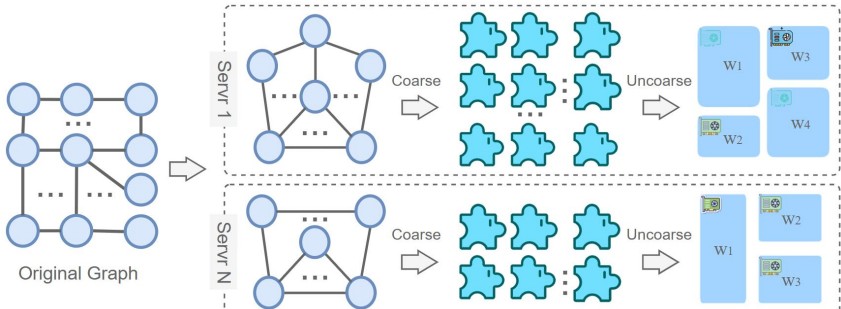

Figure 3: HsysGNN assigns workloads proportional to GPU capability via hierarchical partitioning

Section 3.1. The fraction of the graph assigned to server $s$ is $r_s = \frac{P_s}{\sum_{s' \in \mathcal{S}} P_{s'}}$, where $S$ denotes the set of all servers. This ensures that stronger servers receive proportionally larger workloads.

**GPU-Level Partitioning** Once the graph is partitioned across servers, we further divide each server's subgraph among its local GPUs. For GPU $g$ within server $s$, the workload ratio is computed as $r_g = \frac{P_g}{\sum_{h \in \mathcal{G}_s} P_h}$.

These ratios specify the relative sizes of subgraphs to be assigned to each GPU. To generate concrete subgraphs, we adopt the coarsening–uncoarsening procedure of METIS for uneven partitioning, first collapsing the graph into smaller components and then expanding them into larger partitions with target weights set to the ratios $r_g$. Unlike conventional partitioning, this uneven strategy accounts for GPU heterogeneity by directly embedding compute capability into the partitioning process.

### 3.3 Halo Vertex Optimization

While heterogeneous partitioning balances computational workload, communication overhead remains a major bottleneck due to halo nodes. In full-batch training, halo nodes may even outnumber inner nodes, leading to redundant memory consumption and inflated cross-GPU traffic. To mitigate this problem, HsysGNN employs a halo vertex optimization strategy that prunes low-importance halo nodes while taking bandwidth constraints into account. The importance of a halo node $v$ is measured by its degree. Low-degree nodes make limited contributions and, under bandwidth constraints, can be safely pruned with negligible impact on model accuracy. This bandwidth-aware pruning biases partition boundaries such that halo duplication aligns with high-bandwidth connections, cutting communication overhead without sacrificing balance.

### 3.4 Two-Level CPU–GPU Caching

In heterogeneous clusters, many GPUs lack peer-to-peer (P2P) links, so halo feature transfers must pass through host memory, causing high bandwidth contention when multiple GPUs communicate simultaneously. HsysGNN addresses this with a two-level caching scheme: GPU memory serves as a fast local cache, while CPU memory functions as a larger global cache to bridge non-P2P devices. Fig. 4 shows the architecture. Before sending data, workers check whether halo features already exist in the cache. Cached features are reused or prefetched into GPU memory, thereby reducing redundant communication.

To improve hit rates, we prioritize vertices by their overlap ratio, $R(v_k) = \sum_{i=1}^{P} \mathbb{I}(v_k \in H(G_i))$, where $R(v_k)$ counts the number of partitions containing vertex $v_k$, $P$ is the total number of partitions, $H(G_i)$ is the halo set of partition, $\mathbb{I}(\cdot)$ is the indicator function. High-overlap vertices are cached preferentially, since they are repeatedly accessed across GPUs.

### 3.5 Pipeline-Parallel Execution

Even with optimized partitioning and caching, mismatches between communication latency and GPU compute speed can cause idle time, especially in heterogeneous environments where device

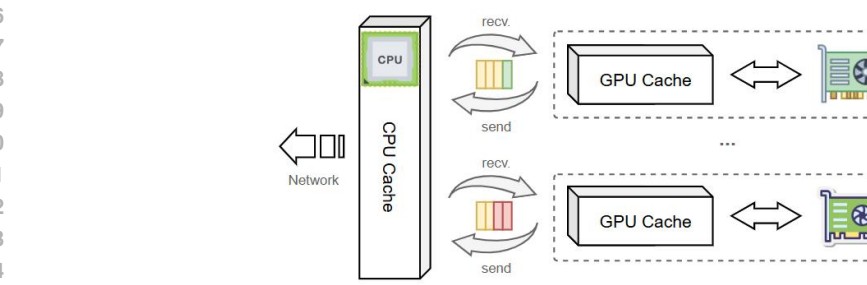

Figure 4: Two-level caching architecture in HsysGNN.

capabilities vary. To mitigate this, HsysGNN employs a pipeline-parallel execution scheme that overlaps data transfers with computation.

We organize data movement into three asynchronous queues: (1) a **local queue** that fetches halo features from the global cache into GPU memory, (2) a **global queue** that aggregates updates into CPU memory for reuse by other GPUs, and (3) a **prefetch queue** that proactively loads features likely needed in the next iteration.

To further reduce synchronization cost, cache updates are performed with lightweight optimistic locks instead of heavy mutexes. This may introduce mild staleness, but we prove in the appendix that training convergence remains unaffected.

## 4 EXPERIMENTS

To evaluate the effectiveness of the proposed HsysGNN, we perform extensive experimental studies using common public datasets and compare HsysGNN with existing methods.

### 4.1 LAB SETUP AND DATASETS

We implement CaPGNN based on DGL 2.3.0 and PyTorch 2.3.0, using Gloo and PCIe 3.0x16 for CPU-GPU communication. Our testbed consists of a heterogeneous GPU cluster with dual Intel® Xeon® Gold 6230 CPUs, 768 GB RAM, and a mixture of GPUs including Tesla A40, RTX 3090, and GTX 3060 Ti. The GPU parameters are detailed in Table 1.

Table 1: Detailed information about NVIDIA GPUs.

| Name | Label | SM Count | Arch. | CUDA Cores | Tensor Cores | FP32 (TFLOPS) | Memory (GB) | Bandwidth (GB/s) |
|---|---|---|---|---|---|---|---|---|
| **NVIDIA RTX 3090** | R9 | 82 | Ampere | 10496 | 328 | 35.58 | 24 | 936.2 |
| **NVIDIA Tesla A40** | T4 | 84 | Ampere | 10752 | 336 | 37.42 | 48 | 695.8 |
| **NVIDIA RTX 3060** | R6 | 28 | Ampere | 3584 | 112 | 12.74 | 12 | 360.0 |
| **NVIDIA RTX 2060 SUPER** | R2 | 34 | Turing | 2176 | 272 | 7.181 | 8 | 448 |
| **NVIDIA GTX 1660Ti** | G6 | 24 | Turing | 1536 | 96 | 5.43 | 6 | 288.0 |

We evaluate HsysGNN on widely used benchmark datasets from OGBN and DGL, spanning small to large scales. Table 2 summarizes their statistics. The accuracy is reported for single-label tasks and F1 score for multi-label ones. We perform the evaluation on GCN Kipf & Welling (2016). Comparisons include the following baselines: (i) Vanilla training without optimization, (ii) state-of-the-art distributed GNN training systems AdaQP Wan et al. (2023), which is more advanced than many traditional methods, such as SANCUS and PipeGCN. Unless specified otherwise, models use three layers, hidden dimension of 256, and learning rate of 0.01, trained for 200 epochs. All methods are tuned with their recommended hyperparameters for fairness.

### 4.2 OVERALL PERFORMANCE

We first compare the end-to-end training performance of HSYSGNN with representative distributed GNN systems in heterogeneous GPU environments. The primary objective is to examine whether

Table 2: Graph datasets used in experiments.

| Dataset | Label | #Nodes | #Edges | #Features | #Classes |
|---|---|---|---|---|---|
| **Reddit** Hamilton et al. (2017) | Rt | 232,965 | 114,615,892 | 602 | 41 |
| **Yelp** Zeng et al. (2020) | Yp | 716,847 | 13,954,819 | 300 | 100 |
| **AmazonProducts** Zeng et al. (2020) | As | 1,569,960 | 264,339,468 | 200 | 107 |
| **ogbn-products** Hu et al. (2020) | Os | 2,449,029 | 61,859,140 | 100 | 47 |

Table 3: Performance comparison across datasets with different partitions, servers, and methods.

| Dataset | Partitions | Server & GPUs M1 | M2 | Method | Throughput (epoch/s) | Accuracy(%) | Dataset | Partitions | Server & GPUs M1 | M2 | Method | Throughput (epoch/s) | Accuracy(%) |
|---|---|---|---|---|---|---|---|---|---|---|---|---|---|
| Rt | 2M-2D | R6(x2) | R9(x2) | Vanilla | 0.10±0.01 | 94.42±0.02 | Yp | 2M-2D | R6(x2) | R9(x2) | Vanilla | 0.08±0.00 | 31.29±0.02 |
| | | | | AdaQP | 0.37±0.02 | **94.60±0.03** | | | | | AdaQP | 0.30±0.00 | 28.99±0.01 |
| | | | | **HsysGNN** | **3.66±0.01** | 94.08±0.01 | | | | | **HsysGNN** | **3.82±0.01** | **42.12±0.22** |
| Rt | 2M-4D | R9(x1) T4(x1) R6(x1) | R9(x2) R2(x2) | Vanilla | 0.08±0.01 | 94.44±0.01 | Yp | 2M-4D | R9(x1) T4(x1) R6(x1) | R9(x2) R2(x2) | Vanilla | 0.09±0.00 | 31.60±0.11 |
| | | | | AdaQP | 0.27±0.01 | **94.48±0.02** | | | | | AdaQP | 0.30±0.01 | 29.41±0.08 |
| | | | | **HsysGNN** | **4.89±0.06** | 94.01±0.03 | | | | | **HsysGNN** | **3.97±0.03** | **42.61±0.44** |
| Os | 2M-2D | G6(x1) R6(x2) | R9(x2) | Vanilla | 0.07±0.00 | 88.39±0.04 | As | 2M-2D | G6(x1) R6(x2) | R9(x2) | Vanilla | 0.05±0.00 | 21.64±0.05 |
| | | | | AdaQP | OOM | | | | | | AdaQP | OOM | |
| | | | | **HsysGNN** | **1.73±0.00** | **90.90±0.12** | | | | | **HsysGNN** | **1.56±0.02** | **51.22±0.15** |
| Os | 2M-4D | R9(x1) T4(x1) R6(x1) G6(x1) | R9(x2) R2(x2) | Vanilla | OOM | | As | 2M-4D | R9(x1) T4(x1) R6(x1) G6(x1) | R9(x2) R2(x2) | Vanilla | OOM | |
| | | | | AdaQP | OOM | | | | | | AdaQP | OOM | |
| | | | | **HsysGNN** | **1.76±0.01** | **85.43±0.15** | | | | | **HsysGNN** | **2.79±0.02** | **51.53±0.21** |

our heterogeneity-aware design improves training throughput and efficiency without compromising accuracy. Results are summarized in Table 3.

Across all datasets and GPU combinations, HSYSGNN achieves the shortest training times and the highest throughput. Uniform partitioning suffers from severe straggler effects, as faster GPUs remain idle while waiting for slower ones. Methods that neglect bandwidth heterogeneity also incur high cross-device communication costs, further limiting scalability. By contrast, HSYSGNN incorporates both compute and communication heterogeneity into the partitioning process, prunes redundant halo vertices, and leverages two-level caching to reduce inter-partition traffic. Together, these optimizations yield speedups of up to $18\times$ compared to uniform partitioning and $66\times$ over the best existing baseline. Importantly, test accuracy remains comparable to or slightly better than other methods, since pruning low-importance halo vertices reduces redundancy without degrading model quality.

### 4.3 PARTITIONING EVALUATION

To evaluate the effectiveness of partition method, we visually examine the evolution of subgraph characteristics across multiple iterations under different partition counts ranging from 2 to 6. As shown in Fig. 5, we track the changes in the number of nodes, the number of edges, and the heuristic scores for each subgraph with the number of iterations. The partition method quickly reduces the cost imbalance across subgraphs. The observed convergence of subgraph scores toward a common range confirms that the proposed method effectively balances costs across partitions. Moreover, as the number of partitions increases, the initial imbalance becomes more pronounced, yet the proposed method consistently guides subgraphs toward a tightly clustered cost distribution. This demonstrates the strong scalability and stability of the proposed method under varying partition granularities. These results

### 4.4 ABLATION STUDY

To disentangle the contributions of individual components in HsysGNN, we conduct an ablation study by progressively enabling caching, heterogeneity-aware partitioning, and their combination. Experiments are performed under two configurations: two machines with two GPUs each (2M–2D) and two machines with four GPUs each (2M–4D). We report throughput in epochs per second and test accuracy in Table 4.

The results show that both caching and heterogeneity-aware partitioning individually bring significant performance improvements over the vanilla baseline. Caching achieves up to $35\times$ speedup by reducing redundant halo communication, while maintaining accuracy. Partitioning alone greatly increases throughput, especially in the 2M–4D setting, but can harm accuracy if applied without

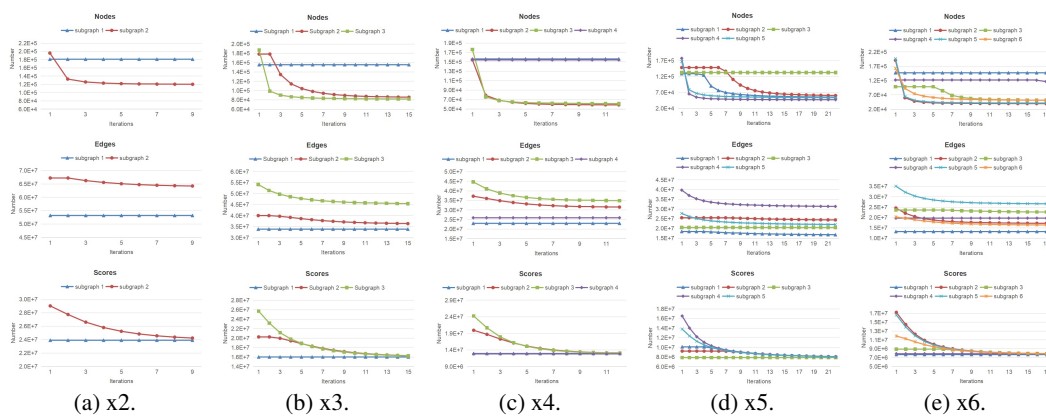

(a) x2.      (b) x3.      (c) x4.      (d) x5.      (e) x6.

Figure 5: Changes in the number of nodes, edges, and scores for each subgraph during partition.

Table 4: Ablation study of HSYSGNN. "+Cache" denotes enabling two-level CPU–GPU caching, "+Partition" denotes enabling heterogeneity-aware partitioning, and "+All" combines both with pipeline execution. Results are averaged over 5 runs.

| Setting | Method | Throughput (epoch/s) | Accuracy (%) |
|---------|--------|----------------------|--------------|
| 2M–2D | Vanilla | $0.10 \pm 0.01$ | $94.42 \pm 0.02$ |
| | +Cache | $3.54 \pm 0.02$ | $93.75 \pm 0.05$ |
| | +Partition | $2.99 \pm 0.01$ | $92.37 \pm 0.01$ |
| | +All | $3.66 \pm 0.01$ | $94.08 \pm 0.01$ |
| 2M–4D | Vanilla | $0.08 \pm 0.01$ | $94.44 \pm 0.01$ |
| | +Cache | $3.41 \pm 0.02$ | $94.21 \pm 0.04$ |
| | +Partition | $5.80 \pm 0.02$ | $90.97 \pm 0.13$ |
| | +All | $4.89 \pm 0.06$ | $94.01 \pm 0.03$ |

halo optimization, as seen from the $3.5\%$ drop. When combined, caching and partitioning achieve the best trade-off, recovering accuracy while sustaining high throughput. This demonstrates that the full design of HsysGNN, including halo-aware caching and balanced partitioning, is necessary to achieve both efficiency and accuracy.

## 5 RELATED WORK

Distributed GNNs, which leverage multi-machine computation and storage, have become a research focus. NeuGraph Ma et al. (2019) integrates deep learning with graph processing, while ROC Jia et al. (2020) improves scalability through graph partitioning and memory optimizations. SAN-CUS Peng et al. (2022) reduces communication overhead through historical embeddings in full-graph training. NeutronStar Wang et al. (2022) combines cached and communicated dependencies using a cost model, and AdaQP Wan et al. (2023) minimizes inter-worker communication via random message quantization, but overlooks halo node impacts. PaGraph Lin et al. (2020) supports sample-based GNN training with caching but struggles with large-scale graphs due to static caching limitations. BGL Liu et al. (2023) improves caching with a FIFO strategy and proximity-sensitive ordering. Legion Sun et al. (2023) optimizes NVLink architectures, and GNNLab Yang et al. (2022) improves parallelism by splitting training into Sampler and Trainer roles. For partitioning, PowerGraph Gonzalez et al. (2012) uses edge-based partitioning to distribute edges evenly, PowerLyra Chen et al. (2019) adopts a hybrid strategy based on vertex degree, and METIS Karypis & Kumar (1998) employs a three-stage process: Coarsening, Initial Partitioning, and Uncoarsening.

## 6 CONCLUSION

In this work, we presented HsysGNN, a distributed GNN training framework designed for heterogeneous GPU clusters. By modeling the hardware environment as a computation–communication

topology graph, HsysGNN adaptively partitions workloads according to device capability and interconnect bandwidth. We further introduced halo vertex optimization to reduce redundant communication and a two-level CPU–GPU caching mechanism with pipeline-parallel execution to hide data transfer latency. Extensive experiments across diverse datasets and heterogeneous GPU combinations demonstrate that HsysGNN significantly accelerates training, up to 66 times compared to the baseline, while maintaining model accuracy. Our work opens several promising directions. One is to extend heterogeneity-aware partitioning to hybrid CPU–GPU or GPU–accelerator systems, where device characteristics are even more diverse. Another is to integrate dynamic profiling so that partitioning can adapt to runtime variability such as thermal throttling or network congestion. Finally, incorporating sampling-based training and more advanced caching strategies may further expand the applicability of HSYSGNN to extremely large-scale graphs.

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

## A   APPENDIX

### A.1   THE USE OF LARGE LANGUAGE MODELS

We used OpenAI's ChatGPT (GPT-5) as a general-purpose writing assistant. Its role was limited to polishing grammar, improving readability, and rephrasing sentences for clarity and conciseness. All ideas, experiments, analyses, and conclusions were conceived, conducted, and validated entirely by the authors. The model did not generate novel research ideas, perform data analysis, or contribute substantively to the scientific content of the paper.

### A.2   REPRODUCIBILITY

We have made every effort to ensure the reproducibility of our results. We will release the source code, scripts for data preprocessing, and instructions for reproducing all results (tables and figures) upon acceptance.

### A.3   PROOF OF CONVERGENCE

We provide a rigorous convergence analysis of GNN training under cache staleness. By incorporating staleness-induced errors into the gradient descent framework, we derive theoretical bounds on gradient norms and demonstrate that convergence can still be achieved under bounded-error conditions. Specifically, we first prove that the embedding error is bounded when using stale embeddings, then demonstrate that the gradient error for each layer is also bounded, finally, establish the global convergence of the training process.

**Lemma A.1.** *Let the infinity norms of matrices $\mathbf{A}$ and $\mathbf{B}$ be defined as $\|\mathbf{A}\|_\infty = \max_{i,j} |\mathbf{A}_{i,j}|$ and $\|\mathbf{B}\|_\infty = \max_{i,j} |\mathbf{B}_{i,j}|$. The following inequalities are satisfied Xue et al. (2023): (a) $\|\mathbf{AB}\|_\infty \leq col(\mathbf{A})\|\mathbf{A}\|_\infty\|\mathbf{B}\|_\infty$, (b) $\|\mathbf{A} \odot \mathbf{B}\|_\infty \leq \|\mathbf{A}\|_\infty\|\mathbf{B}\|_\infty$, and (c) $\|\mathbf{A} + \mathbf{B}\|_\infty \leq \|\mathbf{A}\|_\infty + \|\mathbf{B}\|_\infty$. The $col(\mathbf{A})$ represents the number of columns in matrix $A$, and $\odot$ is the element-wise product.*

**Lemma A.2.** *For $p$ workers, if: (a) the activation function $\sigma(\cdot)$ is $\rho$-Lipschitz continuous, (b) the related matrices $\hat{\mathbf{A}}_i$, $\hat{\mathbf{H}}_i$, $H_i$, and $W_i$ are bounded, (c) the difference between the historical embedding $\hat{\mathbf{H}}_i^{(\ell)}$ and the exact embedding $\mathbf{H}_i^{(\ell)}$ satisfies a staleness bound, i.e. $\|\mathbf{H}_i^{(\ell)} - \hat{\mathbf{H}}_i^{(\ell)}\|_\infty \leq \epsilon_H$, then the approximation error of the intermediate result $\tilde{\mathbf{Z}}_i^{(\ell)}$ is also bounded and satisfies $\|\tilde{\mathbf{Z}}_i^{(\ell)} - \mathbf{Z}_i^{(\ell)}\|_\infty \leq \eta^2 \beta^2 \epsilon_H$. Here, $\epsilon_H$ represents the staleness bound, and $\ell \in [1, L]$ denotes the $\ell$-th layer, $\eta$ is the maximum number of columns that exists in the proof. In addition, $\beta$ is the constant that: $\|\hat{\mathbf{A}}_i\|_\infty \leq \beta, \|\hat{\mathbf{H}}_i\|_\infty \leq \beta, \|\mathbf{H}_i\|_\infty \leq \beta, \|\mathbf{W}_i\|_\infty \leq \beta$.*

*Proof.* In the $i$-th worker, for the GCN model, the forward propagation rule of the $\ell$-th layer is given by:

$$\mathbf{Z}_i^{(\ell)} = \hat{\mathbf{A}}_i \mathbf{H}_i^{(\ell-1)} \mathbf{W}_i^{(\ell)}, \quad \mathbf{H}_i^{(\ell)} = \sigma(\mathbf{Z}_i^{(\ell)}), \tag{3}$$

where, $\hat{\mathbf{A}}_i$ represents the adjacency matrix after symmetric normalization. If a caching algorithm is adopted, the cached intermediate result is $\tilde{\mathbf{Z}}_i^{(\ell)}$, and the forward propagation is given by:

$$\tilde{\mathbf{Z}}_i^{(\ell)} = \hat{\mathbf{A}}_i \tilde{\mathbf{H}}_i^{(\ell-1)} \mathbf{W}_i^{(\ell)}, \quad \tilde{\mathbf{H}}_i^{(\ell)} = \sigma(\tilde{\mathbf{Z}}_i^{(\ell)}). \tag{4}$$

According to Lemma A.1, we have:

$$\begin{aligned}
\|\tilde{\mathbf{Z}}_i^{(\ell)} - \mathbf{Z}_i^{(\ell)}\|_\infty &= \|\hat{\mathbf{A}}_i \tilde{\mathbf{H}}_i^{(\ell-1)} \mathbf{W}_i^{(\ell-1)} - \hat{\mathbf{A}}_i \mathbf{H}_i^{(\ell-1)} \mathbf{W}_i^{(\ell-1)}\|_\infty \\
&\leq \eta \|\hat{\mathbf{A}}_i \tilde{\mathbf{H}}_i^{(\ell-1)} - \hat{\mathbf{A}}_i \mathbf{H}_i^{(\ell-1)}\|_\infty \|\mathbf{W}_i^{(\ell-1)}\|_\infty \\
&\leq \eta^2 \|\hat{\mathbf{A}}_i\|_\infty \|\tilde{\mathbf{H}}_i^{(\ell-1)} - \mathbf{H}_i^{(\ell-1)}\|_\infty \|\mathbf{W}_i^{(\ell-1)}\|_\infty \\
&\leq \eta^2 \beta^2 \epsilon_H.
\end{aligned} \tag{5}$$

$\square$

Next, we will prove that if cached values are used during the forward propagation phase, the error of the model weight gradients obtained in the backward propagation phase is also bounded.

**Lemma A.3.** *We first make the following assumptions. In the $i$-th worker, (a) the activation function $\sigma(\cdot)$ and the gradient $\nabla \mathcal{L}$ are $\rho$-Lipschitz continuous, (b) the matrices $\hat{A}_i$, $W_i$, $\delta_i^{(\ell)}$ and $\sigma'(Z_i)$ are bounded. Thus, we have $\|\nabla_{\tilde{\mathbf{Z}}_i^{(\ell)}} \tilde{\mathcal{L}} - \nabla_{\mathbf{Z}_i^{(\ell)}} \mathcal{L}\|_\infty \leq \rho \eta^2 \beta^2 \epsilon_H$. Here, $\delta_i^{(\ell)}$ denotes the exact gradients, $\sigma'(\mathbf{Z}_i)$ represents the derivative of $\sigma(\mathbf{Z}_i)$ with respect to the input $\mathbf{Z}_i$, $\ell \in [1, L]$ represents the $\ell$-th layer, and $\nabla_{\mathbf{Z}_i^{(\ell)}} \mathcal{L}$ is the gradient matrix of loss $\mathcal{L}$ with respect to $\mathbf{Z}_i^{(\ell)}$ at the $\ell$-th layer.*

*Proof.* The last layer of a GNN is typically directly used for the computation of the loss function, and its gradient depends only on the output of that layer. As a result, the error can be directly controlled using the $\rho$-Lipschitz continuity property. Given that the activation function $\sigma(\cdot)$ and the gradient $\nabla \mathcal{L}$ are $\rho$-Lipschitz continuous, and based on Lemma 1 and Lemma 2, for the $L$-th layer, we have:

$$\|\nabla_{\tilde{\mathbf{Z}}_i^{(L)}} \tilde{\mathcal{L}} - \nabla_{\mathbf{Z}_i^{(L)}} \mathcal{L}\|_\infty \leq \rho \|\tilde{\mathbf{Z}}_i^{(L)} - \mathbf{Z}_i^{(L)}\|_\infty \leq \rho \eta^2 \beta^2 \epsilon_H. \tag{6}$$

According to the characteristics of GNNs, the gradient computation at $\ell$-th layer depends on the results of $(\ell + 1)$-th layer. Additionally, assume that $\|\nabla_{\tilde{\mathbf{Z}}_i^{(\ell')}} \tilde{\mathcal{L}} - \nabla_{\mathbf{Z}_i^{(\ell')}} \mathcal{L}\|_\infty \leq K^{(\ell')}, \forall \ell' > \ell$. Therefore, using the induction hypothesis, the gradient error at $\ell$-th layer can be expressed as:

$$\delta_i^{(\ell)} = \nabla_{\mathbf{Z}_i^{(\ell)}} \mathcal{L} = \frac{\partial \mathcal{L}}{\partial \mathbf{Z}_i^{(l)}} = \delta_i^{(\ell+1)} \hat{\mathbf{A}}_i \left(\mathbf{W}_i^{(\ell)}\right)^\top \odot \sigma'\left(\mathbf{Z}_i^{(\ell)}\right), \tag{7}$$

$$\|\nabla_{\tilde{\mathbf{Z}}_i^{(\ell)}}\tilde{\mathcal{L}} - \nabla_{\mathbf{Z}_i^{(\ell)}}\mathcal{L}\|_\infty = \|\tilde{\delta}_i^{(\ell)} - \delta_i^{(\ell)}\|_\infty$$

$$= \|\tilde{\delta}_i^{(\ell+1)}\hat{\mathbf{A}}_i\big(\mathbf{W}_i^{(\ell)}\big)^\top \odot \sigma'(\tilde{\mathbf{Z}}_i^{(\ell)})$$

$$- \delta_i^{(\ell+1)}\hat{\mathbf{A}}_i\big(\mathbf{W}_i^{(\ell)}\big)^\top \odot \sigma'(\mathbf{Z}_i^{(\ell)})\|_\infty$$

$$\leq \eta^2 \Big\{\|\tilde{\delta}_i^{(\ell+1)}\|_\infty \|\hat{\mathbf{A}}_i\|_\infty \|\big(\mathbf{W}_i^{(\ell)}\big)^\top\|_\infty \|\sigma'(\tilde{\mathbf{Z}}_i^{(\ell)}) - \sigma'(\mathbf{Z}_i^{(\ell)})\|_\infty \qquad (8)$$

$$+ \|\tilde{\delta}_i^{(\ell+1)} - \delta_i^{(\ell+1)}\|_\infty \|\hat{\mathbf{A}}_i\|_\infty \|\big(\mathbf{W}_i^{(\ell)}\big)^\top\|_\infty \|\sigma'(\mathbf{Z}_i^{(\ell)})\|_\infty\Big\}$$

$$\leq \eta^2\Big(\beta^3\rho(\eta^2\beta^2\epsilon_H) + K_{(\ell+1)}\beta^3\Big)$$

$$= \eta^2\beta^3\big(K_{(L)} + K_{(\ell+1)}\big).$$

Define the error bound for $\ell$-th layer as $K_{(\ell)} = \eta^2\beta^3\big(K_{(L)} + K_{(\ell+1)}\big)$. By induction, starting from the final layer $L$ and iterating backward, it can be shown that the gradient error for all layers is bounded, and the error bounds accumulate progressively across layers. This establishes that the training process retains theoretical convergence even in the presence of stale embeddings. $\qquad\square$

Next, we will prove the convergence guarantee of GNN training with cached embeddings under bounded staleness conditions.

**Theorem A.4.** *For a model with $L$ layers, let $\mathbf{W}_t$ denote the model parameters at the $t$-th training iteration, and $\mathbf{W}_\star$ represent the optimal model weights. Assume the following conditions are satisfied: (a) the activation function $\sigma(\cdot)$ and the gradient of loss $\nabla\mathcal{L}$ are $\rho$-Lipschitz, (b) the matrices $\hat{\mathbf{A}}$, $\mathbf{W}$, $\mathbf{H}$, and the gradients of the loss are bounded by the constant $\gamma$: $\|\hat{\mathbf{A}}\|_\infty \leq \gamma, \|\mathbf{W}\|_\infty \leq \gamma, \|\mathbf{H}\|_\infty \leq \gamma, \|\nabla_{\mathbf{W}}\tilde{\mathcal{L}}\|_\infty \leq \gamma, \|\nabla_{\mathbf{W}}\mathcal{L}\|_\infty \leq \gamma, \|\mathcal{L}(\mathbf{W})\|_\infty \leq \gamma$, (c) the loss $\mathcal{L}(\mathbf{W})$ is $\rho$-smooth. Then, there exists $\alpha > 0$, s.t., $\forall T > L\epsilon_H$, if the GNN is trained in parallel with bounded staleness for at most $R \leq T$ iterations, where $R$ is chosen uniformly from $T$ and the learning rate is defined as $\tau = \min\left\{\frac{1}{\rho}, \frac{1}{\sqrt{T}}\right\}$. Thus, the the following bound holds:*

$$\mathbb{E}_R\left[\|\nabla\mathcal{L}(\mathbf{W}_R)\|_F^2\right] \leq \frac{2\left(\mathcal{L}(\mathbf{W}_1) - \mathcal{L}(\mathbf{W}_\star)\right)}{\sqrt{T}} + \frac{\rho\alpha}{2\sqrt{T}}, \qquad (9)$$

*where, $\|\cdot\|_F$ is the Frobenius norm, $T$ is the total epochs.*

*Proof.* Let $\delta_t = \nabla_{\mathbf{W}_t}\tilde{\mathcal{L}} - \nabla\mathcal{L}(\mathbf{W}_t)$ denote the differences between gradients at epoch $t$. By the $\rho$-smoothness of $\mathcal{L}(\mathbf{W})$, the model update rule $\mathbf{W}_{t+1} = \mathbf{W}_t - \tau\nabla\tilde{\mathcal{L}}(\mathbf{W}_t)$, and Lemma A.1, we have:

$$\mathcal{L}(\mathbf{W}_{t+1}) \leq \mathcal{L}(\mathbf{W}_t) + \langle\nabla\mathcal{L}(\mathbf{W}_t), \mathbf{W}_{t+1} - \mathbf{W}_t\rangle + \frac{\rho}{2}\|\mathbf{W}_{t+1} - \mathbf{W}_t\|_F^2,$$

$$= \mathcal{L}(\mathbf{W}_t) + \langle\nabla\mathcal{L}(\mathbf{W}_t), \mathbf{W}_{t+1} - \mathbf{W}_t\rangle + \frac{\rho}{2}\tau^2\|\nabla_{\mathbf{W}_t}\tilde{\mathcal{L}}\|_F^2$$

$$= \mathcal{L}(\mathbf{W}_t) - \tau\langle\nabla\mathcal{L}(\mathbf{W}_t), \nabla_{\mathbf{W}_t}\tilde{\mathcal{L}}\rangle + \frac{\rho}{2}\tau^2\|\nabla_{\mathbf{W}_t}\tilde{\mathcal{L}}\|_F^2$$

$$= \mathcal{L}(\mathbf{W}_t) - \tau\langle\nabla\mathcal{L}(\mathbf{W}_t), \delta_t\rangle - \tau\|\nabla\mathcal{L}(\mathbf{W}_t)\|_F^2$$

$$+ \frac{\rho}{2}\tau^2\big(\|\delta_t\|_F^2 + \|\nabla\mathcal{L}(\mathbf{W}_t)\|_F^2 + 2\langle\delta_t, \nabla\mathcal{L}(\mathbf{W}_t)\rangle\big)$$

$$\leq \mathcal{L}(\mathbf{W}_t) - \left(\tau - \frac{\rho}{2}\tau^2\right)\|\nabla\mathcal{L}(\mathbf{W}_t)\|_F^2 + \frac{\rho}{2}\tau^2\|\delta_t\|_F^2 \qquad (10)$$

$$\leq \mathcal{L}(\mathbf{W}_t) - \left(\tau - \frac{\rho}{2}\tau^2\right)\|\nabla\mathcal{L}(\mathbf{W}_t)\|_F^2$$

$$+ \frac{\rho}{2}\tau^2(\|\nabla_{\mathbf{W}_t}\tilde{\mathcal{L}}\|_\infty + \|\nabla\mathcal{L}(\mathbf{W}_t)\|_\infty)$$

$$\leq \mathcal{L}(\mathbf{W}_t) - \left(\tau - \frac{\rho}{2}\tau^2\right)\|\nabla\mathcal{L}(\mathbf{W}_t)\|_F^2 + \frac{\rho}{2}\tau^2(2\gamma^2)$$

$$\leq \mathcal{L}(\mathbf{W}_t) - \left(\tau - \frac{\rho}{2}\tau^2\right)\|\nabla\mathcal{L}(\mathbf{W}_t)\|_F^2 + \frac{\rho}{2}\tau^2\alpha,$$

where, $\langle \cdot, \cdot \rangle$ denotes the inner product. Furthermore, we have:

$$\forall t, \quad \left(\tau - \frac{\rho}{2}\tau^2\right) \sum_{i=1}^{T} \|\nabla\mathcal{L}(\mathbf{W}_t)\|_F^2 \leq \mathcal{L}(\mathbf{W}_1) - \mathcal{L}(\mathbf{W}_\star) + \frac{\rho}{2}\tau^2\alpha T. \tag{11}$$

According to $\tau = \min\left\{\frac{1}{\rho}, \frac{1}{\sqrt{N}}\right\}$, we can convert Equation 11 to:

$$\begin{aligned}
\mathbb{E}_R\|\nabla\mathcal{L}(\mathbf{W}_R)\|_F^2 &= \frac{1}{T}\sum_{i=1}^{T} \|\nabla\mathcal{L}(\mathbf{W}_t)\|_F^2 \\
&\leq \frac{\mathcal{L}(\mathbf{W}_1) - \mathcal{L}(\mathbf{W}_\star) + \frac{\rho}{2}\tau^2\alpha T}{T\tau\,(2 - \rho\tau)} \\
&\leq \frac{2\left(\mathcal{L}(\mathbf{W}_1) - \mathcal{L}(\mathbf{W}^\star)\right)}{T\tau} + \rho\tau\alpha \\
&\leq \frac{2\left(\mathcal{L}(\mathbf{W}_1) - \mathcal{L}(\mathbf{W}^\star)\right)}{\sqrt{T}} + \frac{\rho\alpha}{2\sqrt{T}}.
\end{aligned} \tag{12}$$

As $T \to \infty$, we can find that $\mathbb{E}_R\|\nabla\mathcal{L}(\mathbf{W}_R)\|_F^2 \to 0$. The result demonstrates that even with stale embeddings in the parallel GNN training, the optimization process maintains theoretical convergence. Additionally, the error introduced by stale embeddings and randomness, represented by the term $\frac{\rho\alpha}{2\sqrt{T}}$, decreases as $T$ increases, showing that the system's approximation error is bounded and controlled. $\qquad\square$

Note that although the above discussion uses GCN as an example, the convergence proof in the paper also applies to other GNN models, provided that the following conditions are met: (a) the activation function and gradients are Lipschitz continuous; (b) the loss function is smooth; (c) the approximation error introduced by cached embeddings is bounded; (d) the aggregation and update rules ensure unbiased gradients. For instance, in GAT, dynamic weights introduce an additional layer, where these weights are computed based on node features. While this may affect gradient computation and propagation, the convergence proof remains valid as long as these computations are Lipschitz continuous and the smoothness conditions hold.

