# OpenReview forum: "HsysGNN: Optimizing Distributed Training of Graph Neural Networks in Heterogeneous Systems"
_ICLR.cc/2026/Conference — ICLR 2026 Conference Withdrawn Submission_

### Official Review · Reviewer_HHpR · 2025-10-25

**Soundness:** 2
**Presentation:** 2
**Contribution:** 2
**Rating:** 2
**Confidence:** 3

**Summary:**

This paper presents HsysGNN, a distributed GNN training framework designed specifically for heterogeneous GPU environments. The authors address the inefficiencies that arise when existing GNN frameworks, which assume homogeneous hardware, are deployed on clusters with mixed GPU capabilities. HsysGNN models the hardware as a computation-communication topology graph to guide workload partitioning proportional to device capabilities. The system incorporates several optimizations including heterogeneity-aware graph partitioning, halo vertex optimization, two-level CPU-GPU caching, and pipeline-parallel execution to improve training performance while maintaining model accuracy.

**Strengths:**

* The paper tackles a practical and increasingly common scenario - heterogeneous GPU clusters resulting from incremental upgrades, cost constraints, or mixed-use environments.
* The approach is holistic, considering both computational and communication heterogeneity through the computation-communication topology graph abstraction.
* The two-level caching strategy and pipeline-parallel execution effectively address communication bottlenecks in non-P2P GPU environments.

**Weaknesses:**

* The paper primarily compares against AdaQP and vanilla training. Comparisons with more recent distributed GNN systems would strengthen the evaluation.
* Experiments are limited to relatively small configurations (2 machines with 2-4 GPUs each). The scalability to larger clusters with dozens or hundreds of GPUs is unclear.
* Experiments focus mainly on GCN with limited evaluation on other GNN architectures (GAT, GraphSAGE mentioned but not thoroughly evaluated).
* The paper doesn't discuss the overhead of profiling GPU capabilities, computing partition ratios, or maintaining the two-level cache system.
* The system appears to use static partitioning based on initial profiling. How it handles dynamic changes in GPU performance (thermal throttling, competing workloads) is not addressed.
* The two-level caching strategy's memory consumption and its impact on available memory for model training is not analyzed.

**Questions:**

* What is the time and computational overhead of profiling GPU capabilities and measuring interconnect bandwidths? How often does this need to be repeated?
* Have you tested HsysGNN on larger clusters (e.g., 8+ machines, 32+ GPUs)? How does the performance scale?
* How does the system handle runtime variations in GPU performance due to thermal throttling or competing workloads? Can the partitioning be adjusted dynamically?
* What is the memory overhead of the two-level caching system? How does this impact the maximum graph size that can be trained?
* How does the quality of heterogeneity-aware partitioning (in terms of edge cuts) compare to standard METIS partitioning? Does prioritizing load balance significantly increase communication?
* Can you provide more comprehensive results for other GNN architectures beyond GCN, particularly attention-based models like GAT?
* How does HsysGNN handle GPU failures during training? Is there any checkpointing or recovery mechanism?
* While you prove convergence under bounded staleness, what is the practical impact on convergence speed? How many additional epochs are typically needed?
* How does your full-batch approach compare with sampling-based methods in heterogeneous environments?

---

### Official Review · Reviewer_wudT · 2025-10-30

**Soundness:** 3
**Presentation:** 2
**Contribution:** 3
**Rating:** 2
**Confidence:** 5

**Summary:**

This paper proposes HsysGNN, a distributed GNN training framework designed for heterogeneous GPU environments. The method models GPUs and interconnects as a computation–communication topology graph to guide workload partitioning, and introduces a two-level CPU–GPU caching strategy combined with pipeline-parallel execution to overlap communication and computation. Experiments show significant throughput improvement over baselines such as AdaQP, while maintaining comparable accuracy.

**Strengths:**

1. The paper addresses a practical and relevant problem: distributed GNN training in heterogeneous clusters, which is becoming increasingly common in real-world deployments.
2. The idea of modeling hardware heterogeneity via a topology graph that combines compute and bandwidth characteristics is intuitive and useful.

**Weaknesses:**

1. Lack of clarity in caching mechanism. The two-level CPU–GPU caching strategy is vaguely described. It is unclear how cache replacement is handled (e.g. LRU/FIFO/priority-based), or how consistency is maintained when cached halo features become stale. The prefetching mechanism is mentioned, but there is no algorithmic detail on how prefetch decisions are made —whether prediction is based on access patterns, iteration history, or static topology.
2. Unclear explanation of overlap mechanism. The pipeline-parallel execution claims to overlap computation with communication, but the paper lacks detailed analysis or timing breakdown to show actual overlap or latency hiding. It would strengthen the work to include timeline diagrams or GPU trace analyses showing concurrent data transfer and computation.
3. Missing cache-related evaluation. Since caching is one of the core contributions, no metric such as cache hit/miss rate, bandwidth utilization, or CPU–GPU traffic reduction is reported. Without these measurements, it is difficult to attribute the performance gains specifically to the caching or prefetching components.
4. Fig. 5 is unclear and contain low-resolution visuals that do not provide useful insight into behavior.

**Questions:**

1. Could you clarify the cache replacement policy and eviction criteria?
2. How are prefetch candidates selected —are they predicted based on historical halo access or statically determined?
3. How is computation–communication overlap validated? Did you measure actual concurrent kernel execution or DMA utilization?
4. Can you provide cache hit/miss statistics or communication volume reduction numbers?
5. Are there scalability results with more servers or larger datasets?

---

### Official Review · Reviewer_uraH · 2025-10-31

**Soundness:** 1
**Presentation:** 1
**Contribution:** 2
**Rating:** 2
**Confidence:** 4

**Summary:**

This paper presents a GNN training method designed to address challenges of training in hetergeneous GPU environment. The authors propose an elegant approach that models hardwares as a computational-communication topology graph, where node features represent computational capability and edge features represent bandwidth. This topology is then used to guide graph partitioning, and optimize communication and computation efficiency by prioritizing important vertices in graph. While the research is well-motivated and the approach is interesting, this paper suffers from limited technical clarity and poor readability. Overall, this paper states a potentially impactful idea to fill an important research gap. However, the presentation currently lacks sufficient technical clarity and polish to allow readers to fully understand or reproduce the method.

**Strengths:**

Strengths:
1. Relevant motivation: This paper targets an important and practical research problem: heterogeneous GPU environment
2. Novel hardware modelling: The modelling of hardware resources as graph is a clever approach which aligns with the graph learning setting

**Weaknesses:**

Weakness:
1. Weak connection between sections: It is unclear how the computational-communication topology graph introduced the Section 3.1 contributes to the subsequent graph partitioning, a clearer explanation linking the two would strengthen the technical coherence of the paper

2. Lack of notation details: The paper lacks explanation for key components, such as
    - how they obtain the model-specific operator weights $w\_{alpha}$ and $w\_{beta}$ in Equation 2
    - The term halo nodes is introduced but never defined
    - Notations are inconsistent and confusing (e.g. $S$ denoting the set of all servers in Section 3.2 is different from $\mathcal{S}$ used in the equation of $r_s$)

3. Poor Readability and presentation quality:
    - Unfinished sentence (‘’These results” in Section 4.3)
    - Typos in Figure 3 (should be ‘Server’ but ‘Servr’)

**Questions:**

Could you please address the issues listed in the weaknesses?

---

### Official Review · Reviewer_kuhQ · 2025-10-31

**Soundness:** 2
**Presentation:** 2
**Contribution:** 2
**Rating:** 2
**Confidence:** 3

**Summary:**

This paper addresses the problems of load imbalance and communication inefficiency in GNN training on heterogeneous distributed GPU clusters.
The authors propose a graph partitioning method that enables their system to partition the graph according to each GPU’s compute capacity and the interconnect bandwidth between GPUs.
In addition to graph partitioning, the authors also propose techniques such as halo vertex optimization, a two-level CPU-GPU cache, and pipeline parallelism.

**Strengths:**

1. It addresses a realistic problem of GNN training on heterogeneous GPU clusters by considering the performance differences in computation and communication across GPUs.
2. It provides a quantitative analysis of the problem rather than relying on intuitive reasoning.
3. Upon the basic solution, the authors propose a series of performance optimization techniques.

**Weaknesses:**

1. There are issues in the Methodology section. For example, Ti in Equation (1) is never mentioned again later, and the paper does not explain how the values of alpha i and beta i in Equation (1) are determined.
2. Some parts only present results without providing insightful analysis. For instance, it is unclear why Equations (1) and (2) are designed this way, and whether they only account for computational differences, while variations in GPU memory bandwidth can also affect SpMM performance. Moreover, the topology graph shown in Figure 2 is never utilized in the subsequent analysis.
3. The role of interconnect bandwidth in graph partitioning is not discussed. For example, line 145 mentions Bij, but its influence on graph partitioning is never explained later.
4. The experimental section lacks comparisons with recent state-of-the-art systems, such as XGNN [VLDB’24].

**Questions:**

1. In Table 4, under the 2M-4D setting, why does combining Cache with Partition result in lower throughput than using Partition alone?
2. The paper initially claims to propose a computation-communication topology-aware partition scheme, yet the role of communication in the partitioning process is never discussed.

---

### Note · Authors · 2025-11-25

I have read and agree with the venue's withdrawal policy on behalf of myself and my co-authors.